# Bioanalysis of the Ex Vivo Labile PACE4 Inhibitory Peptide Ac-[d-Leu]LLLRVK-Amba in Whole Blood Using Ultra-Performance Liquid Chromatography-Tandem Mass Spectrometry Quantification

**DOI:** 10.3390/pharmaceutics15122745

**Published:** 2023-12-08

**Authors:** Max Sauter, Jonas Haag, Cindy Bay, Florian Leuschner, Walter E. Haefeli, Tim Christian Kuhn, Jürgen Burhenne

**Affiliations:** 1Department of Clinical Pharmacology and Pharmacoepidemiology, Heidelberg University Hospital, Im Neuenheimer Feld 410, 69120 Heidelberg, Germany; cindy.bay@med.uni-heidelberg.de (C.B.); walter-emil.haefeli@med.uni-heidelberg.de (W.E.H.); juergen.burhenne@med.uni-heidelberg.de (J.B.); 2Department of Cardiology, Angiology and Pneumology, Heidelberg University Hospital, Im Neuenheimer Feld 410, 69120 Heidelberg, Germany; jonashaag1@gmail.com (J.H.); florian.leuschner@med.uni-heidelberg.de (F.L.); timchristian.kuhn@med.uni-heidelberg.de (T.C.K.); 3German Centre for Cardiovascular Research (DZHK), Partner Site Heidelberg/Mannheim, 69120 Heidelberg, Germany

**Keywords:** PACE4, UPLC-MS/MS, micro-sampling, metabolism, bioanalysis, peptide

## Abstract

The calcium-dependent serine endoprotease PACE4 is evaluated as a therapeutic target for prostate cancer. The peptide Ac-[d-Leu]LLLRVK-amba inhibits PACE4 with high affinity and has shown efficacy in preclinical mice xenograft models of prostate cancer. To support in vivo examinations of the potential therapeutic peptide Ac-[d-Leu]LLLRVK-amba, we established a highly sensitive assay for its quantification in mouse whole blood microsamples based on UPLC-MS/MS determination. Ac-[d-Leu]LLLRVK-amba was very labile during sample processing, which was particularly pronounced in plasma. High resolution mass spectrometric investigations of the metabolism/degradation in plasma revealed that no peptide bond hydrolysis generated products were formed, leaving the cause of the observed consumption of the peptide elusive. As a consequence, whole-blood quantification was developed relying on the immediate snap-freezing of blood samples after collection and immediate sample processing after serial thawing to ensure accurate and reliable quantification. The assay was validated according to the applicable recommendations of the FDA and EMA in a range of 10–10,000 ng/mL and applied to determine the pharmacokinetics of Ac-[d-Leu]LLLRVK-amba after intravenous and intraperitoneal administration to mice. Individual pharmacokinetic profiles were assessed using four microsamplings per animal. Intraperitoneal absorption was found to be efficient, demonstrating that this well-manageable route of administration is feasible for preclinical efficacy experiments with Ac-[d-Leu]LLLRVK-amba.

## 1. Introduction

PACE4 is a calcium-dependent serine endoprotease, which has been identified as a potential therapeutic target in prostate cancer [1,2]. Recently, an efficient inhibitory peptide for PACE4 has been reported, Ac-[d-Leu]LLLRVK-amba, which inhibits PACE4 with low nanomolar affinity. Such high affinity was achieved by optimizing the previously discovered multi-Leu peptide by inserting a C-terminal acetylated D-leucine and replacing the N-terminal arginine with a decarboxylated variant, a 4-amidinobenzylamide (amba) moiety [1].

A major challenge in the development of peptide therapeutics is their susceptibility to enzymatic hydrolysis in biological systems. A primary mechanism of enzymatic degradation of small peptides in the systemic circulation is peptidase-mediated hydrolysis, especially of terminal amino acids by exopeptidase activity [3]. Ac-[d-Leu]LLLRVK-amba comprises stability-enhancing modifications at both peptide termini: a d-amino acid at the N-terminus and an amba-moiety as a decarboxylated arginine variant at the C-terminus. Both modifications impair substrate properties for exopeptidases and thus increase ex vivo plasma stability [1]. The efficacy of Ac-[d-Leu]LLLRVK-amba was previously demonstrated in mouse xenograft models of prostate cancer after daily intravenous and intratumoral injection of 2 mg/kg [1]. However, additionally determined intravenous pharmacokinetic characteristics revealed a very fast clearance from the systemic circulation. Therefore, we investigated intraperitoneal injection as an alternative route of administration and estimated its bioavailability. The rationale was that intraperitoneal absorption may be effective for the small peptide Ac-[d-Leu]LLLRVK-amba and likely flatten the pharmacokinetic profile to prolong the time of potentially effective blood concentrations [4]. In addition, intraperitoneal administration is well-manageable and often induces less stress for animals compared to intravenous injection [4].

For this purpose, we developed a UPLC-MS/MS assay for the sensitive quantification of the peptide in mouse blood microsamples. Nowadays, peptide bioanalysis is primarily performed with LC-MS/MS methodologies, especially due to their potential of fast implementation, wide dynamic range, and superior selectivity compared to immune-based quantifications. Peptide bioanalysis by LC-MS/MS methodologies has several inherent challenges. Apart from the distribution of signal intensity across multiply charged precursor ions and across isotopes, collision-induced dissociation (CID) often involves further signal dilution caused by the predominantly unselective dissociation of peptide bonds leading to multiple, equally abundant product ions. In addition, peptide stability in the used biological matrix is a crucial factor to assess during bioanalytical method development, due to abundance of proteolytic enzymes.

LC-MS/MS-based bioanalysis of Ac-[d-Leu]LLLRVK-amba has previously been only described briefly and was established in a limited dynamic range of 25 to 600 ng/mL in mouse plasma samples [1]. Although the measured plasma concentrations generally exceeded the dynamic range of the assay by more than an order of magnitude, the integrity of the necessary dilution procedure was not verified. We developed an assay with a dynamic range of 10 to 10,000 ng/mL and validated it following the applicable recommendations of the FDA and EMA on bioanalytical method validation [5,6], including stability examinations in plasma and whole blood. The assay was applied and found suitable for the investigation of Ac-[d-Leu]LLLRVK-amba blood concentrations after the intravenous and intraperitoneal administration of 4 mg/kg Ac-[d-Leu]LLLRVK-amba in mice.

## 2. Materials and Methods

### 2.1. Pre-Clinical Mouse Study

Animal studies were approved by the regulatory authorities (Regierungspräsidium Karlsruhe of the state of Baden-Württemberg/Germany, 35-9185.81/G-145/21) and fully complied with European and national regulations for the care and use of laboratory animals (2010/63/EU). Eight- to ten-week-old C57Bl/6 female mice (Janvier) were used for the study.

The animals were administered 4 mg/kg Ac-[d-Leu]LLLRVK-amba in 0.100 mL 0.9% sodium chloride (NaCl) solution per mouse. Four animals received the peptide as intraperitoneal injection, and three animals as intravenous bolus injection into the tail vein. Blood samples were taken 20, 40, 60, and 120 min after administration by collecting a drop of blood from puncture of the facial vein into EDTA tubes (KABE Labortechnik, Nümbrecht, Germany). Whole-blood samples were immediately snap-frozen in liquid nitrogen and stored at −80 °C until analysis.

### 2.2. Drugs, Chemicals, Solvents, and Materials

Ac-[d-Leu]LLLRVK-amba (C_51_H_90_N_14_O_8_; 1026,71 g/mol) and the isotopically labeled internal standard (IS) Ac-[d-Leu]LL-[^13^C_6_, ^15^N]-Leu-RVK-amba were obtained from Peptide Specialty Laboratories GmbH (Heidelberg, Germany). Acetonitrile (ACN) and formic acid (FA) were purchased from Biosolve BV (ULC/MS grade; Valkenswaard, The Netherlands). A 0.9% NaCl solution was purchased from BRAUN. UPLC-grade water was produced with an arium^®^ mini ultrapure water system (Sartorius, Göttingen, Germany). Blank mouse plasma (CD-1) was obtained from Innovative Research (Novi, MI, USA). Pooled human plasma (Li-heparin) and whole blood (K-EDTA) was obtained from healthy donors (ethical vote University of Heidelberg No. S-384/2016).

### 2.3. Standard Solutions

Calibration and quality control (QC) spike solutions were prepared in ACN/H_2_O (1/1, *v*/*v*) + 0.1% FA in glass vials. These were produced from stock solution generated from accurately dissolved independent weighings into 2 mL volumetric flasks. Calibration solutions were prepared at concentrations of 8, 24, 80, 240, 800, 1400, and 8000 ng/mL. QC solutions were prepared at concentrations of 8, 24, 3000, and 6000 ng/mL. The IS spike solution was prepared accordingly at a concentration of 400 ng/mL. Solutions were kept at 4 °C.

### 2.4. Sample Preparation

Blood samples for calibration and QC purposes were generated through the addition of 25 µL of the pertinent spike solution to 20 µL of blank blood and the subsequent addition of 25 µL of IS solution. Calibration samples were produced at 10, 30, 100, 300, 1000, 3000, and 10,000 ng/mL, and QC samples were produced at concentrations of 10, 30, 3750, and 7500 ng/mL. For study sample processing, 20 µL of each sample was added to 25 µL of IS solution, and 25 µL of ACN/H_2_O 1/1 + 0.1% FA was added for volume compensation to ensure that sample preparation matches that of study blood samples. For stability reasons, each sample was immediately depleted from proteins by adding 150 µL ACN including 0.1% FA. Samples were processed individually and kept frozen until processing. Subsequently, samples were centrifuged at 13,200× *g* for 5 min. From the extracts, 10 µL was transferred to 400 µL of ACN/H_2_O 1/19 + 0.1% FA in wells of a 96-well collection plate (Waters, Milford, MA, USA).

### 2.5. Plasma and Whole-Blood Stability

To investigate peptide stability, minimally diluted plasma or blood samples were generated through the addition of 10 µL of a spiking solution (50,000 ng/mL) to 490 µL of the pertinent biological matrix, resulting in a sample concentration of 1000 ng/mL. Samples (withdrawal of duplicates of 20 µL) were measured immediately after preparation and again after 1 and 2 h. Stability was evaluated via the comparison of peak area ratios of analyte and IS.

Metabolite identification experiments were performed in minimally diluted mouse plasma at a Ac-[d-Leu]LLLRVK-amba concentration of 20 µg/mL. Minimally diluted plasma samples were incubated at room temperature and analyzed after 0, 0.5, 4, and 16 h. Samples were depleted from proteins via precipitation with a 2-fold excess of ACN + 0.1% FA and diluted with one equivalent of water + 0.1% FA for direct infusion onto the mass spectrometer.

### 2.6. Instrumental Analysis Parameters

The instrumental setup comprised an Acquity UPLC^®^ Classic System coupled to a Xevo TQ-S triple-stage quadrupole mass spectrometer, which was equipped with a Z-spray heated electrospray ionization (ESI) source (Waters, Milford, MA, USA). The UPLC system was equipped with a Waters BEH Premier C18 Peptide column (300 Å, 1.7 μm, 2.1 × 50 mm) maintained at 60 °C. The mobile phase consisted of two eluents: 5% (v) ACN in water with 0.01% FA (eluent A) and ACN with 0.01% FA (eluent B). The flow rate was 0.5 mL/min and the flow was directed to the ion source between 1.0 and 1.6 min after injection and otherwise to the waste. Chromatographic conditions initially consisted of 95% A/5% B and were kept for 0.1 min after injection. Subsequently, conditions were changed to 40% B within 1.4 min. Then, the ratio was set at 2% A/98% B during 0.2 min and maintained for 0.5 min. Initial conditions were restored during 0.1 min. Starting conditions were maintained for 0.2 min and for an additional minute during preparation of the subsequent injection by the Sample Manager. Samples were refrigerated to 10 °C while in the Sample Manger. The injection needle was washed with ACN/water/MeOH (2/1/1, *v*/*v*/*v*) + 1% FA after each injection. In total, the measurement of each sample was completed in 3.5 min. A volume of 20 µL was used for injection. ESI source parameters were manually optimized to a capillary voltage of 1000 V, source temperature of 150 °C, cone gas flow (N_2_) of 150 L/h, desolvation gas flow (N_2_) of 1000 L/h, and desolvation temperature of 600 °C. The integrated IntelliStart procedures of the MassLynx V4.2 system software (Waters, Milford, MA, USA) were used for automated determination of optimized selective reaction monitoring (SRM) parameters for Ac-[d-Leu]LLLRVK-amba and the IS on the Xevo TQ-S, which was operated with CID in positive ion mode using 0.15 mL/min of argon as collision gas. This yielded an optimal cone voltage of 30 V and collision energy of 10 V. The monitored mass transitions were *m/z* 343.3 → 436.9 for Ac-[d-Leu]LLLRVK-amba and *m/z* 345.6 → 440.4 for the IS.

High-resolution mass spectrometric (HRMS) determinations for the investigation of Ac-[d-Leu]LLLRVK-amba metabolism/degradation in plasma were performed on a Xevo G2-XS QTof (Waters, Milford, MA, USA) equipped with a heated Z-spray ESI source. Measurements were performed via direct infusion using the integrated infusion pump with positive ion sensitivity mode.

### 2.7. Validation of the Analytical Methods

The applicable recommendations for bioanalytical method validation of the FDA and EMA were applied to the validation of the assay [5,6]. Accuracy was calculated as the percentage of the ratio of the mean of the determined concentrations and the nominal concentration. The acceptance limit is 100 ± 15%, with the exception of the LLOQ where it is 100 ± 20% Q. The assay precision was calculated as the coefficient of variation of the determined sample concentrations. It needs to be ≤15% in general and ≤20% at the LLOQ. Each validation run included blank and IS controls, seven calibration samples in duplicate determination, and four QC samples (LLOQ and low, mid, and high QC concentrations) in six-fold determination. Recovery of the whole blood extraction process was calculated from peak areas of QC samples divided by the peak areas obtained from blank blood spiked with the respective amount after processing. Assessment of matrix effects was performed by the ratio of peak areas of blank blood samples spiked after processing with the peak areas of UPLC solvent containing an identical amount of analyte [7]. Stability of the analyte was investigated in freeze-and-thaw cycles separated by at least 24 h and at room temperature for testing bench-top stability.

### 2.8. Calculations and Statistical Methods

Calibration curves were determined with 1/x^2^ weighted linear regression using peak area ratios of the analyte to IS. This calculation was performed using the software TargetLynx V4.2 (Waters, Milford, MA, USA). Blood pharmacokinetics were determined with the software Kinetica (v 5.0; Thermo Fisher Scientific, Waltham, MA, USA). Pharmacokinetic analyses were performed using standard non-compartmental methods. The elimination rate (λ) was calculated by linear regression of ln-transformed concentrations from the terminal concentration decline, and the half-life was calculated as ln 2 divided by λ. Standard calculations were performed using Microsoft Office Excel 2010 (Mountain View, CA, USA).

## 3. Results and Discussion

### 3.1. Mass Spectrometric and Chromatographic Characteristics

Ac-[d-Leu]LLLRVK-amba produced its double- and triple-protonated ions in positive ESI ionization, with the [M+3H]^3+^ ion at *m/z* 343.3 being the most abundant. Because mobile protons facilitate the dissociation of peptide bonds in CID, higher protonation states usually correspond to higher sensitivity in SRM for MS/MS determinations of peptides. In accordance with this principle, the [M+3H]^3+^ ion of Ac-[d-Leu]LLLRVK-amba produced the most intense mass transition to the y_7_-fragment at *m/z* 436.9 (*z* = 2) and was therefore chosen for SRM. The corresponding mass transition of the IS was monitored at identical SRM conditions. A detailed product ion pattern for the [M+3H]^3+^ precursor ion at a low collision energy (10 V) is shown in Figure 1. Apart from b-series and y-series product ions expected to be generated at a low collision energy, several iminium ions are also abundant due to the high number of Leu in the peptide sequence. These include the Leu iminium ion (including d-Leu), the a_1_ product ion corresponding to an acetylated Leu iminium ion, and the dipeptide iminium ion of two Leu (or d-Leu), which includes the three product ion species of deacetylated a_2_, y_7_a_3_, and y_6_a_4_.

Efficient chromatography was achieved with a C18 column with a large pore width and a gradient from 5 to 40% ACN in 1.4 min. The large pores of the column and heating to 60 °C facilitated optimal mass transfer kinetics and produced sharp peaks (width at baseline of 4 s; Figure 2).

### 3.2. Stability of Ac-[d-Leu]LLLRVK-Amba in Mouse Plasma and Fresh Mouse Whole Blood

To determine which biological matrix is best suited for pharmacokinetic experiments in mice, we investigated the bench-top stability of Ac-[d-Leu]LLLRVK-amba in plasma and fresh whole blood at room temperature over a 2 h period (Figure 3). In plasma, anti-coagulated with Li-heparin, Ac-[d-Leu]LLLRVK-amba was very rapidly degraded. The calculated half-life (t_1/2_) was only 0.48 h (29 min). In contrast, the peptide was much more stable in whole blood with a calculated half-life of 1.39 h, but Ac-[d-Leu]LLLRVK-amba was still labile in this biological matrix.

This already very fast degradation at room temperature does not match previous stability investigations in Na-heparin mouse plasma [1], which stated a half-life of 18 h at 37 °C. However, these were conducted at a substantially higher concentration of Ac-[d-Leu]LLLRVK-amba of 500 µg/mL compared to the 1 µg/mL investigated here. This results in vastly different plasma–substrate ratios and, as a likely consequence, in the observed difference in degradation rates. Such lower stability in plasma compared to whole blood was already reported for several peptides [8].

Because enzymatic hydrolysis of the peptide is a potential cause for the instability of Ac-[d-Leu]LLLRVK-amba, we performed an investigation of its metabolism/degradation in plasma at a concentration of 20 µg/mL using HRMS (range of *m/z* 50–1200). This experiment revealed a substantially higher stability of several hours, concordant with the results of the previous report [1]. While the signal for Ac-[d-Leu]LLLRVK-amba substantially decreased during storage in mouse plasma at room temperature over 16 h, no traces of metabolites were observed, neither those formed by peptide bond hydrolysis nor those formed by oxidation. In addition, no other new signals were observed. It is therefore doubtful that chemical degradation or metabolism is the cause of the observed instability of Ac-[d-Leu]LLLRVK-amba in mouse plasma and blood, and the cause for the detected consumption of Ac-[d-Leu]LLLRVK-amba remains elusive. It might be explained by aggregation phenomena or incorporation into plasma components.

Surprisingly, Ac-[d-Leu]LLLRVK-amba was found substantially more stable in human Li-heparin plasma and whole blood with no detectable degradation over at least 2 h, which indicates that the consumption of Ac-[d-Leu]LLLRVK-amba may be species-specific. Because an isotopically labeled analog of Ac-[d-Leu]LLLRVK-amba was used as IS, potential variations in (negligible) matrix effects can be efficiently balanced by the IS due to its identical physicochemical characteristics. As a consequence, whole blood from other species is appropriate for calibration and QC preparation. Therefore, whole blood from human donors was used for these purposes, which also helps to reduce animal sacrifice.

### 3.3. Sample Preparation and Extraction Characteristics

As a consequence of the instability of Ac-[d-Leu]LLLRVK-amba in mouse plasma, accurate concentration measurements in plasma were not possible, and whole blood was chosen as the sample matrix for the pharmacokinetic experiments due to the observed higher stability of Ac-[d-Leu]LLLRVK-amba. Because Ac-[d-Leu]LLLRVK-amba is still relatively labile in whole blood, we snap-froze blood samples immediately after collection and additionally adjusted the sample processing methodology to ensure accurate determinations of Ac-[d-Leu]LLLRVK-amba concentration by avoiding ex vivo incubation in the liquid state of the collected blood samples. For this purpose, each sample was thawed individually and processed immediately, resulting in sequential sample processing. Due to the use of a stable isotopically labeled analog of Ac-[d-Leu]LLLRVK-amba as IS, it can compensate for depletion as soon as it is present in the samples. Therefore, for sample processing, the samples were placed in reaction tubes that already contained the appropriate amount of IS spike solution. The isolation of Ac-[d-Leu]LLLRVK-amba from blood was established via protein precipitation using ACN. This methodology is easily manageable and rapid, and we effectively extracted Ac-[d-Leu]LLLRVK-amba. The immediate addition of ACN for protein precipitation to individually thawed samples ensured the rapid depletion of proteins. While this sequential procedure substantially increases sample throughput time, it facilitates reliable quantification by avoiding analyte instability. The feasibility of this sample processing procedure was confirmed by the stability of Ac-[d-Leu]LLLRVK-amba in the extracts for at least 24 h (deviation found well within ± 15% of nominal concentration; Appendix A).

Due to the high sensitivity of the assay, the obtained extracts had to be further diluted for UPLC-MS/MS quantification. A 40-fold dilution proved to be optimal for the calibrated range. This dilution step enables us to adjust the calibration range of the assay. If required, an even lower LLOQ of 1 ng/mL can easily be achieved by reducing the dilution factor after protein precipitation to five-fold.

The IS-normalized recoveries were well within 15% deviation, with values ranging from 95.1 to 107.2% (Appendix A). Absolute recoveries were not determined due to extensive adsorption of Ac-[d-Leu]LLLRVK-amba to the collection plate in pure mobile phase eluents.

### 3.4. Validation Results

Extraction via protein precipitation combined with UPLC-MS/MS quantification for Ac-[d-Leu]LLLRVK-amba complied with the FDA and EMA recommendations [5,6] for accuracy, precision, linearity, matrix effects, and recovery. Within the calibrated range (10–10,000 ng/mL), correlation coefficients (r^2^) of >0.99 were obtained using linear regression and 1/x^2^ weighting (Appendix A). Table 1 gives an overview of the intraday and interday accuracies and precision values during validation, which were well within the required limits. The observed carry-over after the highest calibration sample was 0.015% and within the required limits of less than 20% of the peak area of the LLOQ. Stock solution stability was established for 2 weeks, which was sufficient to cover sample analyses with an accuracy of 93.7% of the peak areas of the stored solutions compared to initial values.

Reflecting the established sample collection and processing strategy, the stability of Ac-[d-Leu]LLLRVK-amba stability during freeze-and-thaw was solely confirmed for one cycle with determined accuracies of 91.5 to 98.3% (corresponding precision ≤14.0%; Appendix A) while additional freeze-and-thaw did not comply with the stability criteria of 100 ± 15% of initial concentration. Nevertheless, this stability for one freeze-and-thaw cycle is sufficient for the established sample processing methodology. Extracts were stable over the course of individual analyses, as evidenced by accurate reanalysis of low, mid, and high QC samples that remained in the autosampler for 24 h (accuracies between 97.5 and 99.7%, corresponding precision ≤2.5%; Appendix A). This demonstrated extract stability is pivotal for our bioanalytical strategy and further confirms that protein depletion prevents Ac-[d-Leu]LLLRVK-amba depletion.

### 3.5. Matrix Effect

Because Ac-[d-Leu]LLLRVK-amba was extracted with protein precipitation, endogenous peptides and phospholipids can influence the SRM analysis. However, due to the efficient chromatographic separation, no endogenous signals interfered with the analysis of Ac-[d-Leu]LLLRVK-amba (Figure 2). In blank samples from 6 mice, we did not observe interferences at the retention time of the analyte (Appendix A), which demonstrates the specificity of the chosen SRM transition and our assay.

Whole-blood quantification of Ac-[d-Leu]LLLRVK-amba showed negligible absolute matrix effects below −5% for low to high QC samples. This absence of matrix effects is likely caused by the high dilution after sample extraction. Further, the isotopically labeled analog of Ac-[d-Leu]LLLRVK-amba used as the IS efficiently balanced these effects, as demonstrated by IS-normalized matrix effects of 101.1 to 109.1% (Appendix A), which complied well with the required limits.

### 3.6. Pharmacokinetics of Ac-[d-Leu]LLLRVK-Amba in Mice after Intravenous and Intraperitoneal Injection

The pharmacokinetics of Ac-[d-Leu]LLLRVK-amba was already determined after intravenous bolus injection. With a half-life of only 9 min, this route of administration only produced a short period of potentially effective concentrations. This may be the reason why the efficacy of Ac-[d-Leu]LLLRVK-amba in treating prostate cancer in LNCaP xenograft mice was substantially higher when injected intratumorally than when administered intravenously [1], suggesting that prolonged exposure improves efficacy. Therefore, we aimed at flattening the pharmacokinetic profile with alternate routes of administration. For this purpose, we compared intraperitoneal with intravenous injection in mice, due to the good manageability of intraperitoneal administration and the associated reduction in stress for mice during treatment.

The measured pharmacokinetic profiles are shown in Figure 4. As expected, intraperitoneal administration resulted in a considerably flattened profile compared to intravenous bolus injection. The half-life after intravenous injection was comparable to that of intraperitoneal administration with 21.0 ± 1.6 min and 15.9 ± 1.0 min, respectively. The half-life in our experiment was considerably longer than in a previous study [1], which may be due to the fact that we performed later blood samplings and therefore reduce the contribution of the initial distribution phase after intravenous bolus injection. Theoretically, it could also be the result of non-linear pharmacokinetics at the higher dose we used; however, our values in the low concentration ranges show no evidence of non-linear clearances.

Our study has several limitations. Because we demonstrated the specificity of our assay, pre-dose samples were not collected. Due to the late first sampling in our study and the short half-life of Ac-[d-Leu]LLLRVK-amba, c_max_ and t_max_ could not be accurately determined. Further, a large fraction of the area under the concentration–time curve (AUC) (in particular for the intravenous administration) is not captured by our data. As a consequence, the AUC cannot be compared accurately, and the bioavailability of the intraperitoneal administration cannot be calculated. Nevertheless, judging by the determined partial profiles, it can be concluded that intraperitoneal absorption is very efficient.

## 4. Conclusions

We successfully validated a bioanalytical strategy for the pharmacokinetic assessment of the PACE4 inhibitor peptide Ac-[d-Leu]LLLRVK-amba. Due to an elusive depletion of Ac-[d-Leu]LLLRVK-amba, quantification in mouse plasma was not possible. Because the stability of Ac-[d-Leu]LLLRVK-amba in whole blood is considerably higher, pharmacokinetic sampling and determinations in whole blood were preferred. In addition, to minimize the degradation of Ac-[d-Leu]LLLRVK-amba, blood samples were snap-frozen and, after thawing, immediately processed in tubes that already contained the internal standard to balance any occurring degradation. As a consequence of the instability that also occurs in whole blood, samples could only be thawed once without compromising their integrity. Nevertheless, the assay was successfully validated following the guidelines of the FDA and EMA for bioanalytical development. The established bioanalytical strategy enabled the comparison of intravenous and intraperitoneal administration of Ac-[d-Leu]LLLRVK-amba in mice, which revealed that Ac-[d-Leu]LLLRVK-amba is efficiently absorbed after intraperitoneal administration and is suitable as an alternative route of administration to replace intravenous dosing of the peptide in efficacy experiments.

## Figures and Tables

**Figure 1 pharmaceutics-15-02745-f001:**
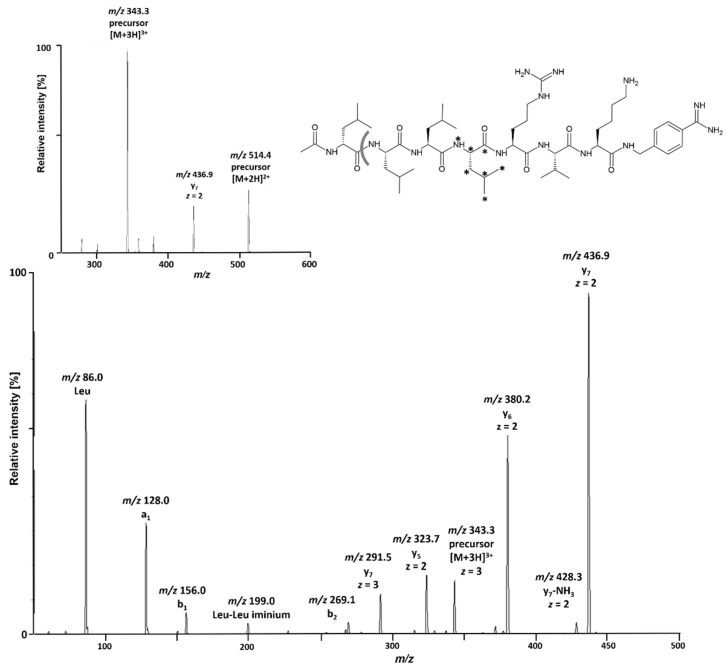
Positive precursor (**top**) and product spectrum (MS/MS) of the [M+3H]^3+^ signal (*m/z* 343.3; **bottom**) of Ac-[d-Leu]LLLRVK-amba with collision-induced dissociation at a collision energy of 10 V. Three letter amino acid labels represent the corresponding iminium ion. The structure of Ac-[d-Leu]LLLRVK-amba depicts the dissociation location of the monitored mass transition (gray line) and the position of the isotopic labels in the internal standard (asterisk).

**Figure 2 pharmaceutics-15-02745-f002:**
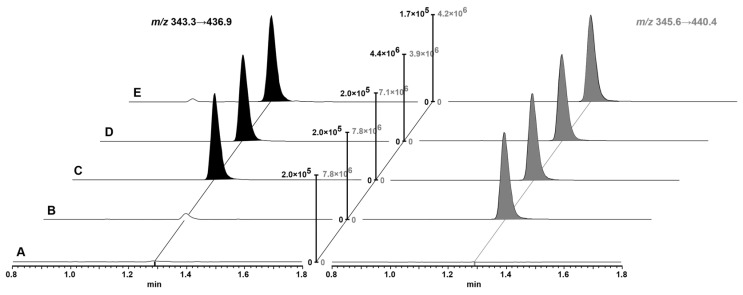
Representative UPLC-MS/MS chromatograms of blood samples of Ac-[d-Leu]LLLRVK-amba. The analyte transition is shown in black and the internal standard (IS) transition in gray. (A) blank sample, (B) sample with added IS, (C) sample at lower limit of quantification (LLOQ) level (representing 10.0 ng/mL), (D) sample at mid QC concentration (representing 3750 ng/mL), and (E) blood sample 2 h after intraperitoneal administration of 4 mg/kg Ac-[d-Leu]LLLRVK-amba to mouse #1 (calculated Ac-[d-Leu]LLLRVK-amba concentration 15.8 ng/mL). The intensity of blanks was normalized to the value of the pertinent peak in the LLOQ chromatogram while intensity in the remaining chromatograms was normalized to the highest peak. IS and analyte transition were processed independently.

**Figure 3 pharmaceutics-15-02745-f003:**
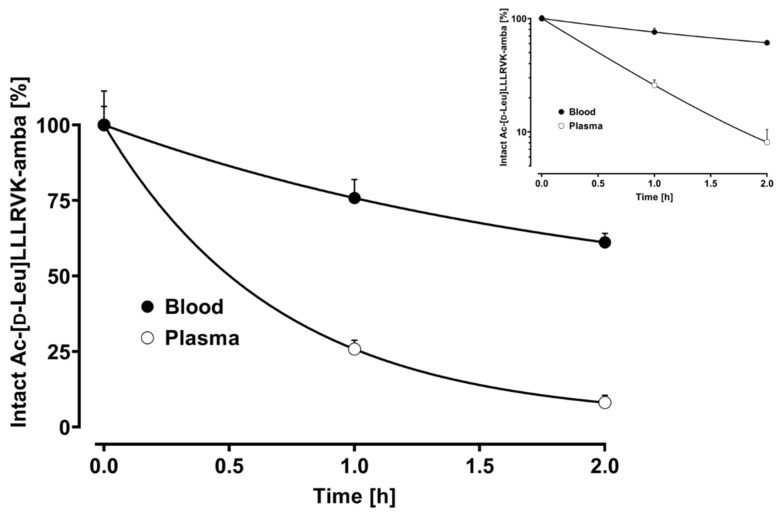
Plasma and whole-blood stability of Ac-[d-Leu]LLLRVK-amba at 1 µg/mL over 2 h at room temperature. The insert shows the corresponding semi-logarithmic presentation.

**Figure 4 pharmaceutics-15-02745-f004:**
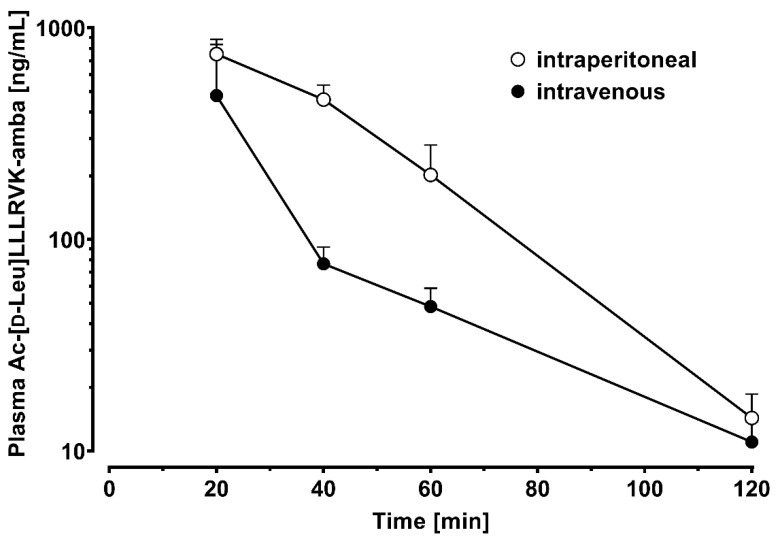
Blood concentration–time profiles of Ac-[d-Leu]LLLRVK-amba after intravenous (*n* = 3) and intraperitoneal (*n* = 4) administration of 4 mg/kg to mice.

**Table 1 pharmaceutics-15-02745-t001:** Quality control results of the assay validation for Ac-[d-Leu]LLLRVK-amba.

		LLOQ	Low QC	Mid QC	High QC
		10.0 ng/mL	30.0 ng/mL	3750 ng/mL	7500 ng/mL
Within-batch					
1	Mean [ng/mL]	9.76	29.8	3981	7357
	Accuracy [%]	97.6	99.2	106.1	98.1
	Precision [%CV]	6.18	5.11	4.76	12.6
2	Mean [ng/mL]	11.6	30.4	4005	8394
	Accuracy [%]	116.3	101.2	106.8	111.9
	Precision [%CV]	1.95	6.49	5.74	2.83
3	Mean [ng/mL]	9.59	26.3	3335	6783
	Accuracy [%]	95.9	87.5	89.0	90.4
	Precision [%CV]	10.9	2.02	2.20	1.48
Batch-to-batch					
	Mean [ng/mL]	10.2	28.7	3762	7407
	Accuracy [%]	102.5	95.8	100.3	98.8
	Precision [%CV]	11.2	8.17	9.68	11.3

CV: coefficient of variation; LLOQ: lower limit of quantification; QC: quality control. *n* = 5 replicates at LLOQ and each QC concentration.

## Data Availability

Data are available from the corresponding author upon reasonable request.

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
