# Peer review of "Bioanalysis of the Ex Vivo Labile PACE4 Inhibitory Peptide Ac-[d-Leu]LLLRVK-Amba in Whole Blood Using Ultra-Performance Liquid Chromatography-Tandem Mass Spectrometry Quantification"

_pharmaceutics, 2023, doi:10.3390/pharmaceutics15122745_

Round 1

Reviewer 1 Report

Comments and Suggestions for Authors

- The paper describes the development and validation of a UPLC-MS/MS assay for the quantification of Ac-[D-Leu]LLLRVK-amba, a PACE4 inhibitory peptide, in mouse whole blood microsamples.

- The paper also investigates the stability and metabolism of the peptide in mouse and human plasma and whole blood, and compares the pharmacokinetics of the peptide after intravenous and intraperitoneal administration in mice.

- The paper is well-written and provides useful information on the bioanalysis and pharmacokinetics of Ac-[D-Leu]LLLRVK-amba, which is a potential therapeutic agent for prostate cancer.

- However, the paper has some major limitations and requires revision before it can be accepted for publication. Some of the main issues are:

  - The authors did not provide sufficient details on the validation of the UPLC-MS/MS assay, such as the calibration curve model, the accuracy and precision at each concentration level, the acceptance criteria, the carry-over effect, etc. The authors should follow the FDA and EMA guidelines more closely and report all the relevant parameters and results in a table format.

  - The authors did not discuss the selectivity or specificity of the assay, which is crucial for peptide bioanalysis. The authors should demonstrate that there is no interference from endogenous or exogenous compounds in the biological matrix, and that the chosen SRM transitions are unique for the analyte and the IS.

  - The authors did not explain the cause of the chemical instability of Ac-[D-Leu]LLLRVK-amba.

Reviewer 2 Report

Comments and Suggestions for Authors

Authors proposed LC-MS/MS based method for the quantification of PACE4 inhibitory peptide in whole blood samples. The method was well developed. as authors reported Levesque et al 2015 briefly discussed about LC-MS methodology in their publication. I have few comments below

1: only advantage of this manuscript is dynamic linearity range.

2: Stability results is not properly addressed. 

3: Peptide hydrolysis: did authors tested at lower temp (ice bath)

4: Please check and confirm calibration and QC values in section 2.3

Reviewer 3 Report

Comments and Suggestions for Authors

The topic of this manuscript is interesting and fits well the scope of pharmaceutics. However, the study design is unclear and the manuscript needs extensive amendments before the final judgement can be made 

1) The authors claim: The assay was validated according to the applicable recommendations of the FDA and EMA in a range of 10-10,000 ng/mL 

Unfortunately they did not do so. Where is the stock solution stability, post-preparative stability and free-thaw cycle stability? If the LLOQ is 8 ng/ml, how come in  a range of 10-10,000 ng/mL ?

2) The animal handling procedures are not disclosed. How did the author collect blood?

3) Why pre-dosing samples were not collected? Without pre-dosing samples, how to demonstrate selectivity?

4) Fig 2 is problematic. Why B has a peak at the retention time of analyte? Why the analyte  signal intensity of C, D and E are similar?

5) Most important, if the analyte is unstable. How to assure the data present in Fig 4 is reliable? The decline in analyte concentration can be due to degradation rather than distribution, metabolism and excretion. 

Round 2

Reviewer 1 Report

Comments and Suggestions for Authors

I have read your revised manuscript and the response letter carefully.

I am pleased to see that you have addressed all the comments from the previous review and improved the quality of your paper.

I have no further comments or suggestions

- I appreciate the authors' efforts to revise their paper according to the previous review comments. They have added the title and reference number of their paper in the response letter, provided detailed point-by-point responses to the reviewers' comments, and made several changes in the paper text, figures, equations, results, discussion, and references.

- The authors have improved the clarity and readability of their paper by correcting some grammatical errors, typos, and formatting issues. They have also provided more details and explanations for some of their methods, results, and conclusions.

Author Response

Thank you very much for your review and comments on the manuscript.

Reviewer 2 Report

Comments and Suggestions for Authors

Thank you for addressing my comments and revising the manuscript.

Author Response

(The authors gave the same response as above.)

Reviewer 3 Report

Comments and Suggestions for Authors

The manuscript has been improved. The reviewer has no objection to accept it.

Author Response

(The authors gave the same response as above.)
